# Which Tyrosine Kinase Inhibitors Should Be Selected as the First-Line Treatment for Chronic Myelogenous Leukemia in Chronic Phase?

**DOI:** 10.3390/cancers13205116

**Published:** 2021-10-12

**Authors:** Takaaki Ono

**Affiliations:** Division of Hematology, Internal Medicine III, Hamamatsu University School of Medicine, 1-2-1, Handayama, Higashi-ku, Hamamatsu 431-3192, Japan; takaono@hama-med.ac.jp; Tel.: +81-53-435-2267/+81-53-434-2910

**Keywords:** chronic myelogenous leukemia, tyrosine kinase inhibitors, imatinib, 2GTKI, nilotinib, dasatinib, bosutinib, treatment-free remission

## Abstract

**Simple Summary:**

This review discusses the optimal selection of BCR-ABL1 tyrosine kinase inhibitors (TKIs) as the first-line treatment for newly diagnosed chronic myelogenous leukemia in chronic phase (CML-CP). With the advent of TKIs, the treatment goals for CML-CP patients have changed from “simply survival” to “survival with adequate quality of life”, hence the number of CML-CP patients aiming to achieve treatment-free remission has increased, irrespective of age or comorbidities. Therefore, optimal selection of TKIs for maximizing the number of patients to achieve treatment-free remission is an important factor for consideration in future studies. To this end, we must understand the advantages and disadvantages of each TKI in terms of treatment response, disease risk at diagnosis, comorbidities, and medical expenses, and use of effective 2GTKIs based on patient background. This review provides insights into “shared decision-making” in individual cases, including the elderly population.

**Abstract:**

With the use of tyrosine kinase inhibitors (TKIs), chronic myelogenous leukemia in chronic phase (CML-CP) has been transformed into a non-fatal chronic disease. Hence, “treatment-free remission (TFR)” has become a possible treatment goal of patients with CML-CP. Currently, four types of TKIs (imatinib, nilotinib, dasatinib, and bosutinib) are used as the first-line treatment for newly diagnosed CML-CP. However, the second-generation TKI (2GTKI), the treatment response of which is faster and deeper than that of imatinib, is not always recommended as the first-line treatment for CML-CP. Factors involved in TKI selection in the first-line treatment of CML-CP include not only patients’ medical background, but also patients’ choice regarding the desired treatment goal (survival or TFR?). Therefore, it is important that clinicians select an appropriate TKI to successfully achieve the desired treatment goal for each patient, while minimizing the development of adverse events. This review compares the pros and cons of using imatinib and 2GTKI for TKI selection as the first-line treatment for CML-CP, mainly considering treatment outcomes, medical history (i.e., desire for pregnancy, aging factor, and comorbidity), and cost. The optimal use of 2GTKIs is also discussed.

## 1. Introduction

Chronic myelogenous leukemia (CML) is a heterogeneous hematopoietic stem cell disease. The Philadelphia (Ph) chromosome, characterized by a chromosome translocation t(9;22) (q22;q11) involving the ABL1 gene on chromosome 9 and BCR gene on chromosome 22, is the primary cause of CML [1]. The BCR–ABL1 fusion gene encodes the active BCR–ABL1 tyrosine kinase (TKI), which promotes cell growth and replication through a downstream signaling pathway [2,3,4,5]. The BCR–ABL1 fusion protein induces disease progression from chronic phase (CP) to blastic phase (BP), with a fatal prognosis [6]. The treatment of CML has changed from chemotherapeutic agents (hydroxyurea and busulfan) since 1959–1982, and interferon-alpha or allogeneic stem cell transplantation in 1983–2000, to BCR–ABL1 tyrosine kinase inhibitors (TKIs), introduced in 2001 [7]. Specifically, the prognosis of chronic myelogenous leukemia in the chronic phase (CML-CP) has improved with the introduction of imatinib, with a 10-year overall survival (OS) of at least 80% in clinical trials [8]. The database of the Timely and Appropriate Registration System for GLIVEC^®^ Therapy (the new TARGET) observational study 1, a prospective cohort study in Japan, recently reported 5-year OS of 90.4% in the imatinib group, 94.4% in the dasatinib group, and 98.4% in the nilotinib group [9]. Deaths due to CML have substantially decreased due to TKI, and the life expectancy is now similar to that of healthy individuals [10]. In this context, the treatment goals for patients with CML-CP are now focused on improving the quality of life and survival while avoiding late complications associated with TKI use. Therefore, the discontinuation of TKI treatment, that is, treatment-free remission (TFR), has attracted attention. The European LeukemiaNet (ELN) guidelines, revised in 2020 (ELN 2020 guideline), described TFR as a new treatment goal for patients with CML-CP who achieved a deep molecular response (DMR) [11]. In addition, the treatment milestones after TKI administration were also changed to achieve a deeper treatment response at an earlier stage [11]. However, only a limited number of patients can achieve TFRs, while approximately 70–80% of patients with CML-CP remain on long-term TKIs.

Since the early 2000s, imatinib has been the only treatment option for TKIs. However, second-generation TKIs (2GTKIs) with increased affinity for BCR–ABL1, as well as ponatinib, a third-generation TKI with high effectiveness for T315I mutations, for which imatinib and 2GTKIs are resistant [12], became available. Owing to these advancements, the mode of treatment could be changed to another TKI agent even when a serious adverse event or a BCR–ABL1 mutation, the main cause of resistance, occurred. Furthermore, the 2GTKI, bosutinib, was approved by the Food and Drug Administration (in December 2017), the European Union (in April 2018), and the relevant authority in Japan (in July 2020) as the first-line treatment for CML-CP. Thus, imatinib and 2GTKIs (nilotinib, dasatinib, and bosutinib) are available as the first-line agents for CML-CP treatment. In the first-line treatment for CML-CP, the response for 2GTKI is faster than imatinib [13,14,15]. In contrast, in previous clinical trials of 2GTKI and imatinib, except for the 5-year progression-free survival (PFS) and OS of the nilotinib 400 mg bid group in the Evaluating Nilotinib Efficacy and Safety in Clinical Trials—newly diagnosed patients (ENESTnd) study [16] (the OS did not differ significantly from the 10-year OS of the imatinib group in the ENESTnd study [17]), PFS and OS were equivalent [15,17,18]. Thus, from the perspective of survival, the advantages of using 2GTKI have not been proven in clinical trials. Despite the limitations of the observational study, the comparison of 2GTKIs (nilotinib and dasatinib) with the imatinib group in the New TARGET observational study 1, a large-scale population-based study in Japan, revealed that OS and PFS in the nilotinib and dasatinib arms were significantly better than those in the imatinib arm [9]. In contrast, 2GTKIs are generally considered to elicit more serious adverse events, including cardiovascular complications, than imatinib. Furthermore, the medical expenses associated with 2GTKI treatment are high. Therefore, imatinib remains the recommended first-line treatment in these aspects [11,19,20,21].

Thus, the type of TKI used in the first-line treatment of CML-CP is influenced by patient and disease backgrounds; hence “shared decision-making” is an effective strategy for patients. This review compares the advantages and disadvantages of using imatinib and 2GTKI from various points of view, and discusses the precautions to be considered for selecting TKIs as the first-line treatment for CML-CP.

## 2. The Impact of Imatinib versus 2GTKI Selection as First-Line Treatment

### 2.1. Disease Risk at Diagnosis

Achieving an optimal treatment response may be challenging if the disease risk of CML at diagnosis is intermediate or high. In the ENESTnd study, the cumulative molecular response (MR) 4.5 achievement rates (BCR–ABL1 international scale (IS) ≤ 0.0032%) for up to 5 years in intermediate- and high-risk cases by the Sokal score were significantly better in the nilotinib than imatinib groups, with 60.4% and 44.9% patients achieving MR4.5 in the nilotinib 300 mg twice-daily (bid) group and 50.0% and 42.3% in the nilotinib 400 mg bid group, and only 32.7% and 23.1% in the imatinib 400 mg once-daily (qd) group [16]. In addition, nilotinib treatment was associated with lower rates of disease progression in patients with intermediate- and high-risk CML-CP [16]. The cumulative major molecular response (MMR, BCR–ABL1 IS ≤ 0.1%) and MR4.5 achievement rates for up to 5 years in the intermediate- and high-risk cases by Hasford score in the dasatinib versus imatinib study in newly diagnosed patients with CML (DASISION) were almost similar (MMR and MR4.5 rate in the intermediate-risk cases: 71% and 43% in the dasatinib 100 mg qd group, and 65% and 28% in the imatinib 400 mg qd group; MMR and MR4.5 rate in the high-risk cases: 67% and 31% in the dasatinib 100 mg qd group, and 54% and 30% in the imatinib 400 mg qd group, respectively) [18]. In the Bosutinib Trial in First-Line Chronic Myelogenous Leukemia (BFORE) study, the MMR rates at 12 months for intermediate- and high-risk Sokal score patients were superior in the bosutinib group, with 45% and 34% in the bosutinib 400 mg qd group and 39% and 17% in the imatinib 400 mg qd group [15]. The above results suggest that the achievement of MMR and MR4.5 of 2GTKI were better than those of imatinib, when the risk of CML disease was high and even intermediate. In contrast, imatinib for patients at low risk is recommended as the first-line treatment in recent clinical guidelines [11,20], since disease progression is less frequent in low-risk CML patients treated with imatinib [18].

The ELN2020 guideline recommends using the European Treatment and Outcome Study (EUTOS) long-term survival (ELTS) score for baseline disease risk [11]. The ELTS score is more accurate in predicting CML-related mortality than the Sokal and EUTOS scores in a validation cohort of 1120 patients treated with imatinib. Recent studies have shown that ELTS scores can accurately predict OS and CML-related mortality in patients with CML-CP treated primarily with imatinib [22]. In a Dutch population-based study, the usefulness of the ELTS score was reported in patients on 2GTKIs, although the difference was not as clear as in the case of imatinib [23]. In this report, although not directly comparable, the 8-year OS of patients with a high-risk ELTS score was as high as 77% in the 2GTKI group, compared with 55% in the imatinib group [23]. Whether 2GTKIs have better survival than imatinib in high-risk ELTS cases requires further investigation.

In a study involving 1151 patients receiving imatinib in the CML Study IV trial, treatment outcomes were compared between patients with or without additional chromosomal aberration (ACA); the so-called major route (second Ph chromosome, trisomy 8, isochromosome 17q, trisomy 19). The 5-year PFS of patients with the major route was 50% (5-year PFS: 90% in patients without ACA) while the 5-year OS in these patients was 53% (5-year OS: 92% in patients without ACA), showing a significant decrease in survival [24]. In addition, the time to achieve complete cytogenetic response (CCyR) and MMR were significantly longer, and there was a significant correlation between disease progression and accelerated phase/blast phase (AP/BP). Therefore, in the ELN2020 guidelines, cases with high-risk ACA (+8, double Ph, i (17q), +19, −7/7q, 11q23, or 3q26.2 aberrations, and complex aberrant karyotypes), including the major route at the time of diagnosis, have a risk evaluation rating of “warning” [11]. In contrast, in the analysis in which the administered TKI contained 2GTKI (patients treated with imatinib: 46.2%, dasatinib: 24.5%, nilotinib: 30.0%, and ponatinib: 8.3%), the presence or absence of ACA did not affect the treatment response and survival rate [25]. However, owing to the small number of cases studied, whether 2GTKI can improve the treatment response and survival rate of patients with high-risk ACA, including the major route, is not clear, and further research with a larger number of cases is required.

### 2.2. Early Molecular Response (EMR) Rate

EMR, especially achieving BCR–ABL1 IS ≤ 10% at 3 months after starting first-line treatment with TKIs, is an important surrogate marker for predicting better PFS and OS irrespective of the type of TKI [13,18,26]. In the New TARGET observational study 1, the OS of the group that achieved EMR was significantly better than that of the group that did not achieve EMR [9]. Furthermore, EMR is a surrogate marker that predicts the achievement of MMR and MR4.5, which is considered as a marker of eligibility for TFR [13,18]. Hence, in the ELN2020 guidelines that placed TFR as one of the new therapeutic goals, if the BCR–ABL1 IS measured within 1–3 months is >10% for patients who did not achieve EMR 3 months after the start of TKI treatment, a change in TKI is recommended [11]. Considering that the criterion for failure in ELN2013 at 3 months was complete hematological response [27], the significance of reaching IS ≤ 10% earlier was emphasized. In contrast, the guidelines ver1.2022 of the National Comprehensive Cancer Network (NCCN) state that even if the IS at 3 months is slightly over 10%, if the IS at 6 months is less than 10%, the case should be considered sensitive to TKI [20].

Table 1 lists the EMR achievement rate (BCR–ABL1 IS ≤ 10% at 3 months) in comparative studies on imatinib and 2GTKIs. Each 2GTKI group showed a significantly better EMR achievement rate than the imatinib group (Table 1) [13,14,15,28]. Furthermore, in the ENESTnd and DASISION studies, the rate of EMR achievement was significantly higher with nilotinib and dasatinib than with imatinib, only in cases with intermediate and high risk of CML disease at diagnosis [13,29].

### 2.3. MMR and MR4.5 Rates

Similar to that in the ELN2013 guideline [27], the optimal response at 12 months after the start of TKI is defined to be more than MMR in the ELN2020 guideline [11]. At the 10-year follow-up of the International Randomized Study of Interferon and STI571(IRIS) trial, the 10-year OS of patients who achieved MMR was significantly better than that of those who did not, while the CML-related mortality was significantly lower [30]. In contrast, in the 3-year follow-up data of the DASSION study, patients who achieved CCyR (BCR–ABL1 IS ≤ 1%) at 12 months in both dasatinib and imatinib groups had significantly better PFS and OS than patients who did not achieve CCyR [14]. However, only among patients who achieved CCyR were PFS and OS comparable, with or without achieving MMR [14]. In addition, if CCyR was achieved in patients with CML-CP using 2GTKI (86 cases of dasatinib and 81 cases of nilotinib) in the initial treatment, long-term event-free survival (EFS) was favorable irrespective of whether MMR was achieved [31]. Thus, whether achieving MMR 12 months after the start of TKI treatment determines the prognosis of CML-CP remains unclear. Therefore, unlike the ELN2020 guideline [11], the NCCN guideline (Version 1.2022), states that if BCR–ABL1 IS ≤ 1% even after 12 months from the start of TKI treatment, the case is considered as TKI sensitive and no change in TKI is recommended [20].

In contrast, early achievement of MMR can be a predictor for the subsequent achievement of MR4.5. An analysis of the German CML study IV using imatinib showed that patients that achieved MMR at 12 and 18 months had a 5-year cumulative MR4.5 achievement rate of 56.7% and 51.1%, respectively. In contrast, MR4.5 achievement rates were only 14.5% and 6% in patients with CCyR but without MMR achievement (0.1% < BCR–ABL1 IS ≤ 1%) at 12 and 18 months, respectively [32]. The time to achieve MMR also predicts a stable maintenance of MR4.5 among patients on imatinib. According to this report, only a few patients were able to maintain stable MR4.5, especially patients who did not achieve MMR after 12 months [33]. The ELN2020 guideline defines TFR as one of the therapeutic goals for CML-CP, and the following note was added: “For patients aiming at TFR, the optimal response (at any time) is BCR–ABL1 IS ≤ 0.01% (MR4) [11]”. According to the NCCN guidelines, the treatment milestone at 12 months was BCR–ABL1 IS ≤ 1%. However, in the ver1.2022 revision, the milestone of the therapeutic effect at 12 months was changed to 0.1% ≤ BCR–ABL1 IS < 1% when long-term survival was the treatment goal, and to BCR–ABL1 IS ≤ 0.1% when TFR was the treatment goal [20]. This clearly shows that when aiming for TFR, the treatment effect target at 12 months after starting TKI treatment should be set as at least BCR–ABL1 IS ≤ 0.1%, especially ≤0.01%. Table 1 lists the MMR and MR4.5 rates in the three clinical trials comparing imatinib and 2GTKIs mentioned above. In these studies, 2GTKIs could achieve MMR and MR4.5 earlier than imatinib, with better achievement rates [15,16,18] (Table 1). Furthermore, in the recently reported 10-year follow-up of the ENESTnd study, the probabilities of achieving the TFR eligibility criteria (the criteria of the ENEST freedom study, i.e., front-line nilotinib therapy for two years or more and DMR for one year or more [34]) in 5 and 10 years were 11.0% and 29.7% in the imatinib group, respectively, 20.9% and 48.6% in the nilotinib 300 mg bid group, respectively, and 20.6% and 47.3% in the nilotinib 400 mg bid group, respectively, showing that nilotinib was significantly better than imatinib [17]. These results suggest that 2GTKI is more advantageous than imatinib in patients with CML-CP who aim for TFR as the therapeutic goal.

### 2.4. Disease Progression

In the ENESTnd study, the two nilotinib groups (300 and 400 mg bid) showed a lower risk of disease progression to AP/BP than the imatinib group [16]. The recently reported 10-year long-term follow-up data from the ENESTnd study also showed that the progression to AP/BP was significantly suppressed in the nilotinib group (9.2% in the imatinib 400 mg qd group, 4.1% in the nilotinib 300 mg bid group, and 2.7% in the nilotinib 400 mg bid group) [17]. Furthermore, even in patients at a high-risk of Sokal score, the cumulative incidence of progression to AP/BP in the imatinib group was higher than that in the nilotinib group (19.1% in the imatinib group, 9.7% in the nilotinib 300 mg bid group and 7.4% in the nilotinib 400 mg bid group). However, in patients with a low-risk of Sokal score, the incidence of progression to AP/BP did not differ among the imatinib group (1.2%), nilotinib 300 mg bid group (2.2%), or nilotinib 400 mg bid group (1.0%) [17]. In the DASISION study, progression to AP/BP by 5 years was comparable between the dasatinib and imatinib groups (4.6% and 7.3%, respectively) [18]. In the BFORE study, the median observation period was as short as approximately 15 months, but progression to AP/BP was 1.6% in the bosutinib group and 2.5% in the imatinib group [15]. The studies could not be compared due to differences in patient backgrounds and disease risk. However, a recent meta-analysis of six large clinical trials that compared imatinib with 2GTKIs showed that disease progression to AP/BP was significantly lower in the case of 2GTKIs than in the case of imatinib [35]. On the basis of the above observations, 2GTKIs suppress progression to AP/BP more than imatinib, and the use of 2GTKIs is advantageous, especially in cases with a high disease risk.

### 2.5. Survival

In the ENESTnd study, the 5-year PFS and OS did not differ significantly between the nilotinib 300 mg bid group and the imatinib 400 mg group [16]; however, the nilotinib 400 mg bid group showed better results than the imatinib 400 mg qd group (PFS: 91% in the imatinib 400 mg group vs. 95.8% in the nilotinib 400 mg bid group, P = 0.02, OS: 91.7% in the imatinib 400 mg group vs. 96.2% in the nilotinib 400 mg bid group, P = 0.03). A recent report of the long-term follow-up data of 10 years of the ENESTnd study showed similar OS and PFS in the imatinib and nilotinib groups (both 300 mg bid and 400 mg bid) [17]. In the DASISION and BFORE studies, the OS and PFS were similar in the imatinib 400 mg, dasatinib 100 mg, and bosutinib 400 mg groups [15,18,28]. In a meta-analysis of the six large clinical trials comparing imatinib and 2GTKI described above, the OS and PFS did not differ significantly between imatinib and 2GTKIs [35]. The advantage of 2GTKI over imatinib in terms of survival has not been demonstrated in large clinical trials. In contrast, despite the bias of observational studies, the new TARGET observational study 1 showed significantly better OS and PFS at 5 and 8 years in the nilotinib and dasatinib groups than in the imatinib group (imatinib group: 5-year OS, 90.4%; 5-year PFS, 89.8%; 8-year OS, 86.2%; 8-year PFS, 85.6%; nilotinib group: 5-year OS, 98.4%; 5-year PFS, 98.4%; 8-year OS, 97.1%; 8-year PFS, 97.1%; dasatinib group: 5-year OS, 94.4%; 5-year PFS, 92.4%; 8-year OS, 94.4%; 8-year PFS, 92.4%) [9]. This may be attributed to 2GTKI possibly having persistent effects without diminishing, even in cases where the dose is reduced due to adverse events. If imatinib is used as the initial treatment for CML-CP, a sufficient amount must be administered at the beginning. In the Japan Adult Leukemia Study Group (JALSG) CML207 study [36], the median daily dose of imatinib for 24 months from the start of treatment was compared among the following three groups: 360 mg qd (defined as the 400 mg group), 270–359 mg qd (defined as the 300 mg group), and less than 270 mg qd (defined as the 200 mg group). The OS and EFS did not differ significantly between the 400 mg qd group and the 300 mg qd group, but they were significantly worse in the 200 mg qd group (400, 300, and 200 mg group with 7-year EFS of 93%, 92%, and 73%, respectively, and 7-year OS of 96%, 96%, and 85%, respectively) [36]. These results show that to obtain a good survival rate with imatinib, a daily dose of at least 300 mg should be maintained for 2 years after initiating the treatment. However, in this study, the imatinib dose was reduced due to adverse effects in approximately 40% of patients, despite a relatively healthy patient background that met the eligibility criteria. On the contrary, reduced dosing of 2GTKIs may result in a similar treatment response and survival as those at the standard dose. Naqvi et al. observed a favorable response and survival rate in patients with CML-CP treated with dasatinib at a daily dose of 50 mg qd, compared with the response in those patients treated with dasatinib at a standard daily dose of 100 mg qd, as shown in the DASISION study [18], although the study design was limited to a single-arm phase 2 trial [37,38]. The lower rates of toxicities, including pleural effusion and interruption due to adverse events, were also observed in a cohort receiving dasatinib at a daily dose of 50 mg qd, compared with those in the classical control of the DASISION study [18]. The treatment approaches, which use a reduced 2GTKI dose as an initial treatment, may minimize treatment-related toxicities and medical costs and ensure the long-term safety of TKI treatment. Future larger-scale studies focused on reduced dosing of other 2GTKIs (nilotinib and bosutinib) and the timing of reduced dosing (at the timing of starting TKI or after achieving molecular response) are warranted.

### 2.6. Age Factor and Comorbidities at Diagnosis of CML

As the advent of TKIs transformed CML-CP into a chronic disease with good prognosis, it has been shown that comorbidities determine the OS of patients with CML-CP rather than the disease itself [39,40]. According to registry data from Western countries, the median age at diagnosis is 60–65 years for patients with CML [41,42], and comorbidities in patients with CML also increase with age [43]. In cases characterized by multiple comorbidities prior to the start of TKIs or in elderly patients, imatinib may be selected because of concerns regarding increased adverse events due to long-term administration of 2GTKIs. However, the above-mentioned new TARGET observational study 1 with real-world data in Japan demonstrated the usefulness of 2GTKI even in such patient groups [40]. In this observational study, 2GTKI was selected in 66.2% of 136 elderly patients aged 65 years and in 55.6% of 99 patients with a comorbidity, as assessed using the Charlson comorbidity index score (CCI score [44]). In particular, even in patients with high CCI score of 4 or higher, the OS was 91.6% when 2GTKI was administered, which was significantly higher than that of the imatinib-administered group (OS, 50%) [40]. In CML patients with comorbidities, it is desirable to aim at TFR as much as possible considering the safety of the long-term administration of any TKI. During the observation period in patients with a CCI score of 4 or higher, the cumulative achievement rate of MR4.5 in the imatinib group was 0%, whereas that in the 2GTKI-administered group was 34.1% in the new TARGET observational study 1. This indicates that good treatment results can be obtained while reducing adverse events by using 2GTKI, even in elderly patients and patients with comorbidities.

As patients with CML have almost normal life expectancy, which has been confirmed in older adults in the TKI era [10,18], it can be expected that the number of patients who desire TFR will increase regardless of age. Although imatinib probably remains an optimal treatment option in elderly patients in aspects of long-term safety profile, the selection of 2GTKIs as the first-line treatment is a considerable strategy for patients with the aim of treatment discontinuation, including elderly patients with mild/moderate comorbidities or without comorbidities. As described in the previous section, the therapeutic approach using reduced dosing of 2GTKI [37,38] plays an important role in the elderly population with or without comorbidities.

### 2.7. Patients Who Wish to Be Pregnant

As pregnancy during TKI consumption is associated with the risk of teratogenicity, it is recommended that young women who wish to become pregnant discontinue TKIs, even if temporarily, from the pre-pregnancy to post-natal period [11,20,45]. To do so, it is necessary to achieve a DMR, which is considered as an eligibility criterion for TKI discontinuation, as soon as possible from the CML diagnosis and aiming for TFR as a therapeutic goal. As mentioned above, 2GTKI was able to achieve MR4.5 earlier and have a higher rate of molecular response than imatinib [15,16,18]. Therefore, 2GTKI allows more patients to aim for TFR earlier. Furthermore, in the 2GTKI discontinuation study, the TFR rate was almost equivalent to that in the imatinib discontinuation study, although the duration of TKI therapy was shorter than that in the imatinib discontinuation study. In the According to Stop Imatinib (A-STIM) study with the re-initiated criterion defined as a loss of MMR [46], the TFR rate was 61% at 36 months with a median period from imatinib initiation to discontinuation of 79 months. In the ENEST freedom study with the re-initiated criterion defined as a loss of MMR [34,47], the TFR rate was 51.6% at 48 weeks and 42.6% at 5-year follow-up, with a median period from nilotinib initiation to discontinuation of 43.5 months. In the first-line dasatinib discontinue (DADI) study with the re-initiated criterion defined as a loss of DMR (BCR–ABL1 IS ≤ 0.0069%) [48], the TFR rate was 55.2% at 1-year follow-up with a median period from dasatinib initiation to discontinuation of 40.2 months. Based on the above reasoning, the guidelines recommend that 2GTKI be selected as the first-line treatment for young female patients with CML-CP who wish to become pregnant, and the advantages of 2GTKI are clear.

### 2.8. Medical Expenses

Although patients who can obtain DMR aim for TFR, approximately 70–80% of patients with CML-CP currently require long-term TKI treatment. As a result, most patients receive TKIs for the rest of their lives. The high cost of TKIs has become a major limitation in terms of patient and national medical economics. Hence, recent reports on various CML-CP treatment models have analyzed the treatment value of 2GTKIs and generic drugs such as imatinib from the perspective of medical expense [49,50,51]. Shih et al. performed a decision analysis to compare the cost effectiveness of generic imatinib and 2GTKI as the first-line treatment for CML-CP [49]. An analysis was performed based on the current standards of the willingness-to-pay threshold in the USA, Europe, and developing countries, in which the TFR rates of both imatinib and 2GTKI were estimated at 50%. In this analysis, compared to imatinib, 2GTKI was not cost effective [49]. The Markov model was not used for this analysis, and changes to 2GTKI, the standard treatment approach when imatinib elicits a suboptimal response, was not set as an analysis model. Therefore, Yamamoto et al. again used the Markov model with a setting that included TFR and evaluated the cost-effectiveness of the four treatment strategies involving imatinib, dasatinib, nilotinib, and the physician’s choice as the first-line treatment for CML-CP at 10 years after the start of TKI treatment [50]. As a result, even with the TFR included in the analysis, first-line treatment with imatinib was cost effective in both the United States and Japan [50]. In a similar study, imatinib was reported as the first-line treatment with high cost effectiveness [51]. In these models, if CCyR is achieved during the first-line treatment with TKI, the same TKI is continued without any changes. However, in practice, there are several cases in which the MMR is set as a milestone for treatment. If MMR is not achieved at 12 months, it is also possible to switch from imatinib to 2GTKI, with the aim of discontinuing TKIs. In contrast, in clinical practice, a relatively large number of patients may be administered low doses of TKI because of adverse events or medical expenses. Reducing the dose of 2GTKI is a noteworthy treatment strategy, as they may be more cost effective than imatinib, including the possibility of subsequent TKI discontinuation. Another issue worth considering is the second attempt at TKI discontinuation in patients who have failed the first attempt at TKI discontinuation. In a recent report, 36% of patients maintained TFR at 36 months after the second attempt at TKI discontinuation [52]. To evaluate the cost effectiveness will require re-evaluation of each TKI using analysis models based on the administration status and requirements in clinical practice.

## 3. What Is the Optimal Use of 2GTKIs in the First-Line Treatment of CML-CP?

### 3.1. Difference in Treatment Response and Survival

To the best of my knowledge, no clinical trial has directly compared 2GTKIs; based on the results of comparative studies between imatinib and each 2GTKI [15,16,17,18,28], the therapeutic effect is equivalent for all 2GTKIs. Under such circumstances, the results of a multicenter randomized phase 3 controlled trial conducted in Japan on nilotinib 300 mg bid and dasatinib 100 mg qd for newly diagnosed CML-CP (JALSG CML212 study) were presented at the 62nd American Society of Hematology Annual Meeting in December 2020 [53]. The cumulative incidence of MR4.5 at 18 months, which was the primary endpoint of this study, was similar in the nilotinib 300 mg bid group (33.0%) and the dasatinib 100 mg qd group (30.8%). The secondary endpoints of PFS and OS were 98.8% and 98.8% in the nilotinib 300 mg bid group and 99.0% and 99.0% in the dasatinib 100 mg qd group, respectively, showing excellent results with no significant difference between the groups [53]. Thus, the treatment response and survival rate of each 2GTKI can be considered equivalent.

### 3.2. Differences in Adverse Events

The adverse event profiles of 2GTKIs differ, owing to the differences in their off-target effects. Nilotinib can impair glucose tolerance and dyslipidemia relatively early after treatment initiation [16]. The incidence of adverse events related to vascular occlusion, especially a cardiovascular event (CVE), has become a problem due to long-term administration. The 5-year results of the ENESTnd study showed that the cumulative CVE incidence of up to 5 years was 3.2% in the imatinib 400 mg qd group, 10.6% in the nilotinib 300 mg bid group, and 17.9% in the nilotinib 400 mg bid group. The rate was significantly higher in the nilotinib group, and dose dependence was observed [16]. The 10-year long-term follow-up data of the ENESTnd study showed that the cumulative CVE incidence of up to 10 years increased to 6.3% in the imatinib 400 mg qd group, 24.8% in the nilotinib 300 mg bid group, and 33.4% in the nilotinib 400 mg bid group [17]. Patients with intermediate- and high-risk of Framingham risk scores [54] showed a high incidence of CVE after 5 years of nilotinib treatment [16]. However, caution is required, considering that patients with low-risk Framingham risk scores also show increased incidence of CVE after 5 years, albeit less frequently [16]. Based on the above results, nilotinib should be avoided as much as possible in patients with a history of or concurrent vascular events, including myocardial infarction, cerebral infarction, and arteriosclerosis obliterans before treatment, as well as in patients with uncontrolled diabetes and dyslipidemia.

When using dasatinib, attention must be paid to pleural effusion and pulmonary arterial hypertension (PAH). According to 5-year follow-up data from the DASISION trial, 28% of patients (3% for grades 3–4) developed pleural effusion [18]. The median time of onset was 10 months, although it was also observed 3 years after administration, indicating that it is a late-onset complication. In the Japanese sub-population cohort of the DASISION study [55], the incidence of pleural effusion was 42%, which was higher than that in the entire cohort. More than half of the patients developed the disease more than 1 year after dasatinib treatment. Age (≥65 years) has been reported as a risk factor for the development of pleural effusion in patients treated with dasatinib as the first-line treatment [56]. PAH has been widely recognized as an infrequent but lethal adverse effect associated with dasatinib, with an estimated frequency of 5% in the DASISION study (0.4% in the imatinib arm) [18] and 0.45% in the French Pulmonary Hypertension registry [57]. The median time from the initiation of dasatinib to the diagnosis of PAH is 31.5 months (1 week–75 months) [58]. Risk factors associated with the development of dasatinib-induced PAH have not been identified because of small numbers of PAH cases. Pleural effusion is reportedly more frequent in patients with dasatinib-induced PAH (68%) [58]. The 5-year follow-up results from the DASISION study show that the incidence of all grades of arterial ischemic disease and CVE in the dasatinib group tended to be higher, at 5% and 4%, compared with 2% and 2% in the imatinib group, respectively [18]. Considering the above results, dasatinib should be avoided in patients with a history of or concurrent pulmonary diseases, especially chronic obstructive pulmonary disease, which could cause PAH. In contrast, while it is not always necessary to avoid dasatinib in patients with a history or concurrence of vascular events, including myocardial infarction, cerebral infarction, and arteriosclerosis obliterans before treatment, careful monitoring is required.

Bosutinib is characterized by the development of gastrointestinal symptoms (mainly diarrhea, nausea, and vomiting) and hepatotoxicity in the early stages of administration [15,28,59]. The incidence of diarrhea was 70.1% in the BFORE study (not including Japanese patients) [15] and 86.7% in the phase 2 study (B1048 study) [59] in Japanese patients with newly diagnosed CML-CP on 400 mg qd bosutinib, although most cases were of grades 1–2, which can be managed with anti-diarrheal agents. Diarrhea appeared at a median of 1 day after bosutinib treatment initiation [15,59]. Hepatotoxicity, particularly elevation in alanine aminotransferase (ALT) level, is the most notable adverse event associated with bosutinib treatment. The incidence of ALT elevation in the BFORE study and the B1048 study was 30.8% and 55%, respectively, showing a higher incidence of hepatotoxicity in Japanese patients [15,59]. The median time to the appearance of ALT elevation in the B1048 study was 15 days [59]. Withdrawal or dose reduction of bosutinib was required based on the severity. Renal dysfunction may also be observed in patients treated with bosutinib; however, most cases are transient [60]. The Bosutinib Efficacy and Safety in Newly Diagnosed Chronic Myeloid Leukemia (BELA) study (comparative analysis of imatinib 400 mg qd and bosutinib 500 mg qd for newly diagnosed CML-CP) showed that the frequency of occurrence of adverse events of arterial ischemic disease, including CVE, was comparable to that observed with imatinib (imatinib 400 mg qd group, 3.59%; bosutinib 500 mg qd group, 4.84%) [61]. The 5-year results of the BFORE study presented at the 62nd American Society of Hematology Annual Meeting in December 2020 show the incidence of vascular events was 7.5% in the bosutinib arm and 3.4% in the imatinib arm [28]. The above results indicate that bosutinib should be avoided when chronic gastrointestinal disease, liver disease, and renal disease are observed. However, at present, the risk of developing adverse events such as vascular occlusion with bosutinib is lower than that observed with other 2GTKIs. Table 2 lists the proposals for TKI selection based on several medical conditions before starting 2GTKI treatment.

## 4. Conclusions and Future Perspectives

TKIs have significantly improved the life expectancy of almost all patients with CML-CP. Understanding the advantages and disadvantages of each TKI in terms of disease risk at diagnosis, presence or absence of comorbidities, medical costs, and use of appropriate 2GTKI based on patient background is therefore essential. Suggestions for TKI selection for first-line treatment in patients with newly diagnosed CML-CP are shown in Figure 1.

The key take-home messages from this review are as follows:For patients with intermediate- or high-risk disease, 2GTKIs are a better first-line treatment option for CML-CP, due to their potency, which induces a faster and deeper response and reduces disease progression compared with imatinib in this population. Imatinib can be considered for patients with low-risk disease.The use of 2GTKIs could increase the number of patients who achieve eligibility for TFR faster than imatinib; hence, they are recommended for patients with the aim of TFR as a treatment goal.There is no survival benefit of 2GTKIs compared to imatinib in clinical trials; however, a sufficient amount of imatinib (≥300 mg qd) must be administered at the beginning to obtain a good survival rate, if imatinib is selected as the first-line treatment for CML-CP.Imatinib remains an optimal treatment option in elderly patients, considering long-term safety. In contrast, selecting 2GTKIs as the first-line treatment is considered for patients with the aim of treatment discontinuation, including elderly patients. The therapeutic approach of using reduced 2GTKI dosing plays an important role in elderly population.The use of 2GTKIs is strongly recommended as a first-line treatment option for female patients with CML-CP who wish to become pregnant.Although 2GTKIs are more costly than imatinib, re-evaluation of the cost effectiveness of each TKI using analysis models based on the administration status and requirements in clinical practice is essential.The therapeutic effect is equivalent for all 2GTKIs; hence, selecting the optimal TKI should be based on the adverse event profiles (Table 2).

The treatment goals of each patient should be shared by clinicians, and the optimal TKI must be individually selected. At present, globally, the average life expectancy of healthy individuals is >70 years, while in Japan, it is >80 years. Since the median age of onset of CML is 60–65 years, the number of patients with CML-CP who desire TFR will increase, irrespective of age or comorbidities. Furthermore, for comorbidities, such as vascular events which increase with age, TFR is beneficial for the elderly population if treatment discontinuation reduces the long-term toxicities associated with TKI. To aim at TFR in a large number of patients, including elderly individuals and those with comorbidities, the appropriate recognition of candidates eligible for TFR also becomes important, in addition to the selection of TKIs (imatinib or 2GTKIs). Therefore, more accurate and sensitive methods for minimal residual disease (MRD) measurement compared to real-time quantitative PCR (RT-qPCR) are required for a better selection of CML patients with TFR possibilities. Recent studies have successfully assessed MRD in CML patients with undetectable MRD using RT-qPCR. Using dPCR methods for evaluating treatment response has been reportedly effective in TKI treatment for CML-CP [62,63]. In addition, the association between the detection of somatic mutations at initial diagnosis by using next-generation sequencing methods and prognosis (resistance and disease progression) has recently been reported [64,65,66]. These latest findings may give us valuable information for the optimal selection of TKI, appropriate monitoring for treatment response, and the recognition of patients eligible for TFR in the future.

## Figures and Tables

**Figure 1 cancers-13-05116-f001:**
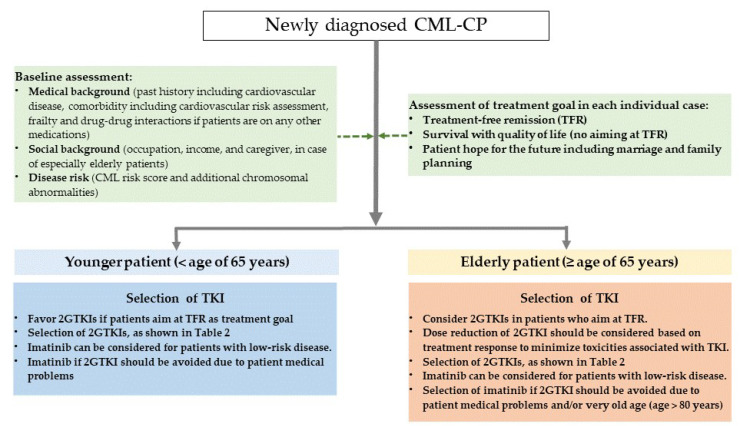
Suggestion for TKI selection for first-line treatment in patients with newly diagnosed CML-CP.

**Table 1 cancers-13-05116-t001:** Clinical outcomes of approved TKIs for first-line treatment of newly diagnosed CML-CP in phase III trials: comparison between imatinib and 2GTKIs.

Study	TKI (Daily Dose)	Number of Patients	BCR-ABL1 ≤ 10% at 3 mo	MMR at 12 mo	MMR by 12 mo	MMR by 5 y	MR4.5 by 5 y	Disease Progression at 5 y	PFS/EFS at 5 y	OS at 5 y
**DASISION**	Dasatinib 100 mg	259	84.0%	N/A	46.0%	76.0%	42.0%	4.6%	85.0%	91.0%
	Imatinib 400 mg	260	64.0%	N/A	28.0%	42.0%	33.0%	7.3%	86.0%	90.0%
**ENESTnd**	Nilotinib 300 mg × 2	282	91.0%	44.0%	55.0%	77.0%	53.5%	3.7%	92.2%	93.7%
	Nilotinib 400 mg × 2	281	89.0%	43.0%	51.0%	77.2%	52.3%	2.2%	95.8%	96.2%
	Imatinib 400 mg	283	67.0%	22.0%	27.0%	60.4%	31.4%	7.9%	91.0%	91.7%
**BFORE** ** ^※^ **	Bosutinib 400 mg	268	75.2%	47.2%	N/A	73.9%	47.4%	2.2%	93.3% ‡	94.5%
	Imatinib 400 mg	268	57.3%	36.9%	N/A	64.6%	36.6%	2.6%	90.7% ‡	94.6%

DASISION, dasatinib versus imatinib study in newly diagnosed patients with CML; ENESTnd, Evaluating Nilotinib Efficacy and Safety in Clinical Trials-newly diagnosed patients; BFORE, Bosutinib Trial in First-Line Chronic Myelogenous Leukemia; TKI, tyrosine kinase inhibitor; CML-CP, chronic myelogenous leukemia in chronic phase; 2GTKI, second generation TKI; mo, month; y, year; MMR, major molecular response; MR, molecular response; PFS, progression-free survival; EFS, event-free survival; OS, overall survival; These data can not be directly compared due to differences in trial methods and patient background. MMR and MR4.5 were defined as BCR-ABL1 international scale ≤ 0.1% and ≤0.0032%, respectively. **^※^** The 5-year results of the BFORE trial were presented in 53rd Amercan Society of Hematology. ‡ event-free survival.

**Table 2 cancers-13-05116-t002:** Selection of TKIs for first-line treatment in patients with newly diagnosed CML-CP based on several comorbidities.

	Imatinib	Nilotinib	Dasatinib	Bosutinib
Comorbidities				
Hypertension (uncontrollable at diagnosis of CML) ^※^				
Diabetes (uncontrollable at diagnosis of CML) ^※^				
Hyperlipidemia (uncontrollable at diagnosis of CML) ^※^				
Cardiovascular disease (including past history)				
Peripheral arterial disease (including ASO)				
Pleural effusion and/or pulmonary disease (including PAH)				
Liver disease				
Renal disease ^‡^				
Gastrointestinal disease				

TKI, tyrosine kinase inhibitor; CML-CP, chronic myelogenous leukemia in chronic phase; ASO, arterio-sclerosis obliterans; PAH, pulmonary arterial hypertension; ^※^ Drug-drug interactions should be focused on for management of these comorbidities. ^‡^ Imatinib and bosutinib have been shown to be safe with closely monitoring in patients with moderate level of renal dysfunction. 

—Preferrable, 
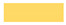
—Controversial, 
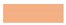
—Less preferable.

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
