# Peer review of "Which Tyrosine Kinase Inhibitors Should Be Selected as the First-Line Treatment for Chronic Myelogenous Leukemia in Chronic Phase?"

_cancers, 2021, doi:10.3390/cancers13205116_

Round 1

Reviewer 1 Report

Overall, this is a well written article summarizing the advantages and disadvantages of TKIs in CML. My only comment is that the article seems a bit biased towards use of 2nd generation TKIs and benefits appear to be overstated. What are the scenarios in which imatinib would be a good first choice? In low risk chronic phase CML patients without any additional cytogenetic abnormalities, imatinib is a perfectly fine first choice.

  1. In section 2.1, it has been stated that 2nd generation TKIs help even in high risk CML patients. The interpretation from these studies however clearly states the benefit is "only" in high risk and maybe some intermediate risk patients. Imatinib is perfectly fine in low risk patients. This should be emphasized better in the text.
  2. Lines 151-153: I would replace "even" with "only" in high and intermediate risk patients. The faster early molecular response with nilotinib and dasatinib over imatinib is not shown in low risk patients.

Reviewer 2 Report

Comments

This manuscript entitled “Which Tyrosine Kinase Inhibitors Should be Selected as the First-line Treatment for Chronic Myelogenous Leukemia in Chronic Phase?” well-summarized previously studies and the clinical view on the 2nd-generation TKI selection for 1st-line treatment and therapeutic strategies of patients with CML in chronic phase. This interesting review is very informative and certainly readable to cancer researchers.

Reviewer 3 Report

Review Takaaki Ono

Which Tyrosine kinase inhibitors should be selected as the first-line treatment for chronic myelogenous leukemia in chronic phase?

The author summarizes the current status of Imatinib compared to second line TKI such as Dasatininb, Nilotinib und the “latest developed” Busotinib. He listed different key studies of CML treatment such as ENESTnd, TARGET and DASISION etc. Different conclusions have been drawn by the scientific community leading to adoption in the 2020 guideline from Europe, USA and also Japan. Main goals that have been emphasized in these studies are not only PFS and OS, which only shows only slight differences among the different drugs. A potential new issue that rises treatment-free remission (TFR) that has gained importance from a clinical but also from a financial point of view. Important knowledge has been gained by molecular data leading to an intensively studied early molecular response (EMR) after treatment. This showed that there is a potential difference among the drugs with deeper response with second line drugs compared to Imatinib however with an increase of cardiac side effects. Also, an advantage of 2GTKI drugs has been reported in elderly patients with/without comorbidities compared to imatinib. Indeed, pregnancy has been evaluated and emphasizes early TFR with 2GTKI but not Imatinib. However, not surprisingly concerning cost effectiveness Imatinib was superior over 2GTKI although TFR could be achieved earlier with 2GTKI than Imatinib.

The author has summarized several relevant clinical trials dealing with all aspects of patients’ treatment with TKI suffering from CML.  The author cited timely up – to date literature.

Minor criticism:

There are several short terms mentioned in the paper that have not been explained. I strongly recommend to provide an abbreviation section or alternatively explain all short terms when shown for the first time.

Reviewer 4 Report

The Review entitled: " Which Tyrosine Kinase Inhibitors Should be Selected as the First-line Treatment for Chronic Myelogenous Leukemia in Chronic Phase? " is an attempt to investigate the advantages and disadvantages of using Imatinib or 2GTKI in CML treatment.

The review is comprehensive, detailed and the references are appropriate and adequate.

Some additional comments:

  • In the introduction, it would be appreciated a brief description of the CML pathology, the clinical features, and progression.
  • A table with the definition of molecular responses level by peripheral blood should be included.
  • In the era of MR4.5 and MR5 which is the best test to detect BCR-ABL? Does real-time quantitative PCR provide sufficient sensitivity? What about digital PCR? A short paragraph on this important issue should be included.
  • The title of paragraph 2 is not clear.
  • In the conclusion section, a take-home message that summarizes all the topics addressed would be clearer.
  • Best regards
